# ADVERSARIAL DEEP METRIC LEARNING

## ABSTRACT

Learning a distance metric between pairs of examples is widely important for various tasks. Deep Metric Learning (DML) utilizes deep neural network architectures to learn semantic feature embeddings where the distance between similar examples is close and dissimilar examples are far. While the underlying neural networks produce good accuracy on naturally occurring samples, they are vulnerable to adversarially-perturbed samples that can reduce their accuracy. To create robust versions of DML models, we introduce a robust training approach. A key challenge is that metric losses are not independent — they depend on all samples in a mini-batch. This sensitivity to samples, if not accounted for, can lead to incorrect robust training. To the best of our knowledge, we are the first to systematically analyze this dependence effect and propose a principled approach for robust training of deep metric learning networks that accounts for the nuances of metric losses. Using experiments on three popular datasets in metric learning, we demonstrate the DML models trained using our techniques display robustness against strong iterative attacks while their performance on unperturbed (natural) samples remains largely unaffected.

## 1 INTRODUCTION

Many machine learning (ML) tasks rely on ranking entities based on the similarities of data points in the same class. Deep Metric Learning (DML) is a useful technique for such tasks, particularly for applications involving test-time inference of classes that are not present during training (e.g., zero-shot learning). Example applications of DML include person re-identification (Hermans et al., 2017), face verification (Schroff et al., 2015; Deng et al., 2019), phishing detection (Abdelnabi et al., 2020), and image retrieval (Wu et al., 2017; Roth et al., 2019). At its core, DML relies on state-of-the-art deep learning techniques that can produce lower-dimensional semantic feature embeddings of high-dimensional inputs. Points in this embedding space cluster similar inputs together while dissimilar inputs are far apart.

Unfortunately, the underlying deep learning models are vulnerable to adversarial examples (Szegedy et al., 2014; Biggio et al., 2013) — inconspicuous input changes that can cause the model to output attacker-desired values. Thus, DML models themselves are vulnerable to adversarial examples. Given their wide usage in diverse ML tasks, including security-oriented ones, it is important to train robust DML models that withstand attacks. This paper tackles the open problem of training DML models using robust optimization techniques (Ben-Tal et al., 2009; Madry et al., 2018).

A key challenge in robust training of DML models concerns the so-called *metric losses* (Wu et al., 2017; Wang et al., 2019; Chechik et al., 2010; Schroff et al., 2015). Unlike loss functions used in typical deep learning settings, DML loss for a single data point depends on the other data points in the mini-batch. A sampling process selects points for a mini-batch, and thus, the DML losses are sensitive to this process as well. For example, the widely-used triplet loss requires three input points: an anchor, a positive sample similar to the anchor, and a negative sample dissimilar to the anchor. To compute this loss, for a batch of size $B$, this would require $O(B^3)$, making the training process inefficient. Thus, a sampling process ensures that a mini-batch contains enough positive and negative examples for the training to be useful while keeping the batch small enough to be efficient.

This *dependence* of the DML loss on the contents of the mini-batch poses a challenge to adversarial training: (1) it is unclear what points should be adversarially perturbed; and (2) it is unknown whether the perturbations would cause training instability. Training a DML model is sensitive to

the sampling process, and selecting samples that are too "hard" or "negative" can lead to training collapse (Wu et al., 2017).

We systematically approach the above challenges and contribute a robust training objective formulation for DML models by considering the widely-used triplet loss. Our key insight is that during an inference-time attack, adversaries seek to perturb data points such that the intra-class distance maximize, and thus this behavior needs to be accounted for during training to improve robustness. Recent work has attempted to train robust DML models, but they do not consider the issue of loss dependence and sensitivity to sampling (Abdelnabi et al., 2020). This leads to non-robust DML models (Panum et al., 2020).

**Contributions.**

- We contribute a principled robust training framework for Deep Metric Learning models by considering the dependence of triplet loss on the other data points in the mini-batch and the sensitivity to sampling.

- We experiment with three commonly-used datasets for vision-based deep metric learning (CUB200-2011, CARS196, SOP) and show that naturally-trained models do not have any robustness — their accuracy drops to close to zero when subjected to PGD attacks that we formulate.

- Using our robust formulation, we achieve good robustness. For example, for a PGD attacker with five iterations and $\|\delta\|_\infty < 0.01$, we obtain an adversarial accuracy of 48.7 compared to the state-of-the-art natural accuracy baseline of 71.8 for the SOP dataset (in terms of R@1 score, a common metric in DML to assess the accuracy of models). Furthermore, the resulting robust model accuracies are largely unaffected for natural (unperturbed) samples.

## 2 RELATED WORK

**Deep Metric Learning.** Deep Metric Learning (DML) is a popular technique to obtain semantic feature embeddings with the property that similar inputs are geometrically close to each other in the embedding space while dissimilar inputs are far apart (Roth et al., 2020). DML employs a variety of *metric losses* such as contrastive (Hadsell et al., 2006), triplet (Schroff et al., 2015), lifted-structure (Hermans et al., 2017), and angular loss (Wang et al., 2017). Recent surveys (Roth et al., 2020; Musgrave et al., 2020) highlight that performance of newer metric losses are lesser than previously reported. Thus, we choose to focus on the two established metric losses, contrastive and triplet loss, as they are widely used and have good performance.

**Adversarial Robustness.** Since early work in the ML community discovered adversarial examples in deep learning models (Szegedy et al., 2014; Biggio et al., 2013), a big focus has been to train adversarially-robust models. We focus on robust optimization-based training that utilizes a saddle-point formulation (min-max) (Ben-Tal et al., 2009; Madry et al., 2018). To the best of our knowledge, no prior work has considered training DML models using robust-optimization techniques. Recent work, however, has used metric losses to improve adversarial training for standard deep network architectures (e.g., CNNs) (Mao et al., 2019; Li et al., 2019). These techniques use metric losses (e.g., triplet) instead of traditional ones (e.g. cross-entropy). By contrast, our goal is to create a robust training objective for DML models themselves. This requires considering the dependence of metric losses on mini-batch items and the sampling process that derives those items. We propose a principled framework for robustly training DML models that considers these factors.

Duan et al. (2018) propose a framework that uses generative models (e.g., GANs (Goodfellow et al., 2014)) during training to generate hard negative samples from easy negatives. We observe that this work is concerned with better *natural* training of DML models rather than adversarial training, which is the focus of this work.

## 3 TOWARDS ROBUST DEEP METRIC MODELS

First, we describe some basic machine learning (ML) notation and concepts required in this paper. We assume a data distribution $\mathcal{D}$ over $\mathcal{X} \times \mathcal{Y}$, where $\mathcal{X}$ is the sample space and $\mathcal{Y} = \{y_1, \cdots, y_L\}$ is the finite space of labels. Let $\mathcal{D}_{\mathcal{X}}$ be the marginal distribution over $\mathcal{X}$ induced by $\mathcal{D}$ [1]. Given $Y \subseteq \mathcal{Y}$ we define $\mathcal{D}_Y$ to be the measure of the subsets of $\mathcal{X} \times Y$ induced by $\mathcal{D}$. For $y \in \mathcal{D}$, $\mathcal{D}_y$ and $\mathcal{D}_{-y}$ denote the measures of the sets $D_{\{y\}}$ and $D_{\mathcal{Y} \setminus \{y\}}$, respectively.

In the *empirical risk minimization (ERM)* framework we wish to solve the following optimization problem:

$$\min_{w \in \mathcal{H}} E_{(\mathbf{x},y) \sim \mathcal{D}} \, l(w, \mathbf{x}, y) \tag{1}$$

In the equation given above $\mathcal{H}$ is the hypothesis space and $l$ is the loss function. We will denote vectors in boldface (e.g. $\mathbf{x}$, $\mathbf{y}$). Since the distribution is usually unknown, a learner solves the following problem over a data set $S = \{(\mathbf{x}_1, y_1), \cdots .(\mathbf{x}_n, y_n)\}$ sampled from the distribution $\mathcal{D}$.

$$\min_{w \in \mathcal{H}} \frac{1}{n} \sum_{i=1}^{n} l(w, \mathbf{x}_i, y_i) \tag{2}$$

Once we have solved the optimization problem given above, we obtain a $w^* \in \mathcal{H}$ which yields a classifier $F \colon \mathcal{X} \to \mathcal{Y}$ (the classifier is usually parameterized by $w^*$, but we will omit it for brevity).

### 3.1 DEEP METRIC MODELS

A *deep metric model* $f_\theta$ is function from $\mathcal{X}$ to $S^d$, where $\theta \in \Theta$ is a parameter and $S^d$ is an unit sphere in $\mathbb{R}^d$ (i.e. $\mathbf{x} \in S^d$ iff $\|\mathbf{x}\|_2 = 1$). Since deep metric models embed a space $\mathcal{X}$ (which can itself be a metric space) in another metric space, we also sometimes refer to them deep embedding. Deep metric models use very different loss functions than typical classification networks described previously. Next we discuss two kinds of loss functions – contrastive and triplet. Let $S = \{(\mathbf{x}_1, y_1), \cdots, (\mathbf{x}_n, y_n)\}$ be a dataset drawn from $\mathcal{D}$. A *contrastive* loss function $l_c$ is defined over a pair $(\mathbf{x}, y)$, $(\mathbf{x}_1, y_1)$ of labeled samples from $\mathcal{X} \times \mathcal{Y}$ and is defined as:

$$l_c(\theta, (\mathbf{x}, y), (\mathbf{x}_1, y_1)) = 1_{y=y_1} \, d_\theta(\mathbf{x}, \mathbf{x}_1) + 1_{y \neq y_1} \, [\alpha - d_\theta(\mathbf{x}, \mathbf{x}_1)] \tag{3}$$

In the equation given above, $1_E$ is an indicator function for event $E$ (1 if event $E$ is true and 0 otherwise), and $d_\theta(\mathbf{x}, \mathbf{x}_1)$ is $\sum_{j=1}^{d}(f_\theta(\mathbf{x})_j - f_\theta(\mathbf{x}_1)_j)$, the $\ell_2$ distance in the embedding space. A *triplet* loss function $l_t$ is defined over three $(\mathbf{x}, y)$, $(\mathbf{x}_1, y_1)$ and $(\mathbf{x}_2, y_2)$ labeled samples and is defined as follows:

$$l_t(\theta, (\mathbf{x}, y), (\mathbf{x}_1, y_1), (\mathbf{x}_2, y_2)) = 1_{y=y_1} \, 1_{y \neq y_2} \, [d_\theta(\mathbf{x}, \mathbf{x}_1) - d_\theta(\mathbf{x}, \mathbf{x}_2) + \alpha]_+ \tag{4}$$

In the equation given above $[x]_+$ is $\max(x, 0)$.

### 3.2 ATTACKS ON DEEP METRIC MODELS

Assume that we have learned a deep embedding network with parameter $\theta \in \Theta$ using one of the loss functions described above. Next we describe how the network is used. Let $A = \{(\mathbf{a}_1, c_1), \cdots, (\mathbf{a}_m, c_m)\}$ be a reference or test dataset (e.g. a set of faces along with their label). $A$ is distinct from the dataset $S$ used during training time. Suppose we have a sample $\mathbf{z}$ and let $k(A, \mathbf{z})$ be the index that corresponds to $\arg\min_{j \in \{1, \cdots, m\}} d_\theta(\mathbf{a}_j, \mathbf{z})^2$. We predict the label of $\mathbf{z}$ as $lb(A, \mathbf{z}) = c_{k(A,\mathbf{z})}$ (we will use the functions $k(.,.)$ and $lb(.,.)$ throughout this section).

Next we describe test-time attacks on a deep embedding with parameter $\theta$. Let $A_y \subset A$ be the subset of the reference dataset with label $y \in \mathcal{Y}$ (i.e. $A_y$ is equal to $\{(\mathbf{a}_j, y) \mid (\mathbf{a}_j, y) \in A\}$). Let $\mathbf{z} \in \mathcal{X}$. *Untargeted attack* on $\mathbf{z}$ can be described as follows (we want the perturbed point to have a different label than before):

$$\begin{aligned} &\min_{\delta \in \mathcal{X}} \; \mu(\delta) \\ &\textit{such that } lb(A, \mathbf{z}) \neq lb(A, \mathbf{z} + \delta) \end{aligned} \tag{5}$$

---

[1] The measure of set $Z \subseteq \mathcal{X}$ under distribution $\mathcal{D}_{\mathcal{X}}$ is the measure of the set $Z \times \mathcal{Y}$ in distribution $\mathcal{D}$.

[2] In case one or more anchors share the minimal distance to $\mathbf{z}$, the tie is broke by a random selection among these anchors.

*Targeted attack* (with a target label $t \neq lb(A, \mathbf{z})$) can be described as follows (we desire to the predicted label of the perturbed point to be a specific label):

$$
\begin{aligned}
&\min_{\delta \in \mathcal{X}} \ \mu(\delta) \\
&such \ that \ lb(A, \mathbf{z} + \delta) = t
\end{aligned}
\tag{6}
$$

In the formulations given above we assume that $\mathcal{X}$ is a metric space with $\mu$ a metric on $\mathcal{X}$ (e.g. $\mathcal{X}$ could $\mathbb{R}^n$ with usual norms, such as $\ell_\infty$, $\ell_1$, or $\ell_p$ (for $p \geq 2$)). Any algorithm that solves the optimization problem described above leads to a specific attack on deep metric models.

### 3.3 ROBUST DEEP METRIC MODELS

Let $S = \{(\mathbf{x}_1, y_1), \cdots, (\mathbf{x}_n, y_n)\}$ be a dataset drawn from distribution $\mathcal{D}$. For a sample $(\mathbf{x}_i, y_i)$ where $1 \leq i \leq n$ we define the following surrogate loss function $\hat{l}(\theta, (x_i, y_i), S)$ for the contrastive loss function $l_c$ :

$$
\hat{l}(\theta, (x_i, y_i), S) = \frac{1}{n} \sum_{j=1}^{n} l_c(\theta, (\mathbf{x}_i, y_i), (\mathbf{x}_j, y_j))
\tag{7}
$$

Similarly, for the triplet loss function $l_t$ we can define the following surrogate loss function:

$$
\hat{l}(\theta, (x_i, y_i), S) = \frac{1}{n_{y_i} n_{y_i}^-} \sum_{j=1}^{n_{y_i}} \sum_{k=1}^{n_{y_i}^-} l_t(\theta, (\mathbf{x}_i, y_i), (\mathbf{x}_j, y_j), (\mathbf{x}_k, y_k))
\tag{8}
$$

Let $S_y$ and $S_{-y}$ be defined as the following sets: $\{(\mathbf{x}, y) \mid (\mathbf{x}, y) \in S\}$ and $\{(\mathbf{x}, y') \mid (\mathbf{x}, y') \in S$ and $y' \neq y\}$. In the equation given above the sizes of the sets $S_y$ and $S_{-y}$ are denoted by $n_y$ and $n_y^-$.

Having defined the surrogate loss function $\hat{l}$ the learner's problem can be defined as:

$$
\min_{\theta \in \Theta} \frac{1}{n} \sum_{i=1}^{n} \hat{l}(\theta, (\mathbf{x}_i, y_i), S)
\tag{9}
$$

Recall that the learner's problem for the usual classification case is:

$$
\min_{w \in \mathcal{H}} \frac{1}{n} \sum_{i=1}^{n} l(w, \mathbf{x}_i, y_i)
\tag{10}
$$

Note that in the classification case the loss function $l(w, \mathbf{x}_i, y_i)$ of a sample $(\mathbf{x}_i, y_i)$ does not depend on the other samples in the dataset $S$. However, in the deep metric model case the surrogate loss function $\hat{l}(\theta, (\mathbf{x}_i, y_i), S)$ for a sample $(\mathbf{x}_i, y_i)$ depends on the rest of the data set $S$. This is the main difference between the embedding and classification scenarios.

*Formulation 1.* Let $B_p(\mathbf{x}, \epsilon)$ denote the $\epsilon$-ball around the sample $\mathbf{x}$ using the $\ell_p$-norm. The straightforward robust formulation is given in the equation below.

$$
\min_{\theta \in \Theta} \max_{(\mathbf{z}_1, \cdots, \mathbf{z}_n) \in \prod_{j=1}^{n} B_p(\mathbf{x}_j, \epsilon)} \frac{1}{n} \sum_{i=1}^{n} \hat{l}(\theta, (\mathbf{z}_i, y_i), S)
\tag{11}
$$

In the formulation given above, all samples are adversarially perturbed at the same time (note that the $\max$ is outside the summation). Therefore, this formulation is not convenient for current training algorithms, such as SGD and ADAM. This is because the entire dataset $S$ has to be perturbed at the same time. Moreover, this formulation is not conducive to various sampling strategies used in training of deep metric models.

*Formulation 2.* In this formulation we push the $\max$ inside the summation sign of the optimization formulation.

$$
\min_{\theta \in \Theta} \frac{1}{n} \sum_{i=1}^{n} \max_{\mathbf{z} \in B_p(\mathbf{x}_i, \epsilon)} \hat{l}(\theta, (\mathbf{z}, y_i), S)
\tag{12}
$$

In the formulation given above, the sample $\mathbf{x}_i$ is adversarially perturbed while other samples in the dataset $S$ are kept intact while computing the surrogate loss function $\hat{l}$. This formulation is more

conducive to current training techniques, such as SGD and ADAM. Moreover, this formulation is also conducive to various strategies for sampling pairs and triplets used in deep metric models.

*Formulation 3.* Our third formulation adds a regularizer which enforces the following informal constraint: if $\mathbf{x}$ changes a bit, the distance in the embedding space does not change too much.

$$\min_{\theta \in \Theta} \frac{1}{n} \sum_{i=1}^{n} [\hat{l}(\theta, (\mathbf{z}, y_i), S) \;+\; \lambda \max_{\mathbf{z} \in B_p(\mathbf{x}_i, \epsilon)} d_\theta(\mathbf{z}, \mathbf{x}_i)] \tag{13}$$

These robust optimization formulations follow the common notion of robustness from robust optimization (Ben-Tal et al., 2009), thus given an algorithm for solving one of the robust optimization formulations, leads to a robust model.

### 3.4 ATTACK ALGORITHM

Performing test-time attacks requires an algorithm to solve the optimization objective described in Equation 5. To evaluate the adversarial robustness of DML models, we propose the attack algorithm seen in Algorithm 1. The intuition of the algorithm is that adversaries seek to push a data point (under attack) $\mathbf{x}_i$ further away from a close data point $\mathbf{x}_j$ of the same label in the embedding space, thus consequently closer to an embedding of data point of a different label. Recall that the embedding space is normalized to the unit sphere. To select a close data point, the algorithm firstly examines the label of nearest neighboring data point in the embedding space $k(A^{(i)}, x_i)$ for the respective set of anchors $A^{(i)}$. If the retrieved label is different from the label of the data point under going attack $y_i$, then no perturbation is performed as the data point is already misclassified under nearest neighbor inference, commonly used within DML. If they share classes, then data point $\mathbf{x}_i$ is adversarial perturbed by approximating a solution to $\arg\max_{\mathbf{z} \in B_p(x_i, \epsilon)} d_\theta(\mathbf{z}, \mathbf{x}_j)$ using established attacks methods, such as Fast Gradient Sign Method (FGSM) (Goodfellow et al., 2015), Carlini-Wagner (C&W) (Carlini & Wagner, 2017), Projected Gradient Decent (PGD) (Madry et al., 2018). The algorithm, and the mentioned attack methods, are applicable to commonly used norms in robust optimization.

---

**Algorithm 1** Evaluation of adversarial robustness for a model parameterized by $\theta$ and a dataset $S = \{(\mathbf{x}_1, y_1), \cdots, (\mathbf{x}_m, y_m)\}$ under some $\epsilon$ for respective $\ell_p$-norm.

---
   **for** $i = 1 \dots m$ **do**
      $A^{(i)} \leftarrow S \setminus \{(x_i, y_i)\}$ *// Define the set of anchors used for inference*
      $j \leftarrow k(A^{(i)}, x_i)$ *// Find the index of the nearest neighbor data point (using $d_\theta(\cdot, \cdot)$)*
      **if** $y_i = y_j$ **then**
         *// Nearest neighbor has the same class as $\mathbf{x}_i$, thereby perform adversarial perturbation*
         $\mathbf{x}_i' \leftarrow \arg\max_{\mathbf{z} \in B_p(x_i, \epsilon)} d_\theta(\mathbf{z}, \mathbf{x}_j)$ *// Approximate using established attack methods*
      **else**
         $\mathbf{x}_i' \leftarrow \mathbf{x}$ *// No perturbation for incorrect predictions*
      **end if**
      **eval**$(\theta, \mathbf{x}_i', y_i, A^{(i)})$ *// Optional step: Perform evaluations*
   **end for**

---

Following the adversarial perturbation, **eval**$(\theta, \mathbf{x}_i', y_i, A^{(i)})$ is an explicit evaluation step, that we use throughout experiments, however it is optional for the actual attack. For more details on evaluations performed throughout the experiments, see Section 4.

### 3.5 ADVERSARIAL TRAINING

To improve robustness of DML models, we propose a training objective that align with the covered robust formulations and can be viewed as adversarial training (Madry et al., 2018) for DML models. Given a set of possible perturbations $\Delta_p = \{\delta \mid \|\delta\|_p \leq \epsilon\}$, parameterized by $\epsilon$ and an $\ell_p$-norm, let $\rho((\mathbf{x}_i, y_i), (\mathbf{x}_j, y_j), (\mathbf{x}_k, y_k))$ be a function that outputs an adversarial perturbation for the data point $\mathbf{x}_i$, with respect to the two dependent datapoints $\mathbf{x}_j$ and $\mathbf{x}_k$, be defined as:

$$\rho((\mathbf{x}_i, y_i), (\mathbf{x}_j, y_j), (\mathbf{x}_k, y_k)) = \arg\max_{\delta \in \Delta_p} 1_{y_i = y_j} d_\theta(\mathbf{x}_i + \delta, \mathbf{x}_j) - 1_{y_i \neq y_k} d_\theta(\mathbf{x}_i + \delta, \mathbf{x}_k) \tag{14}$$

Given a triplet $(\mathbf{x}_i, y_i), (\mathbf{x}_j, y_j), (\mathbf{x}_k, y_k)$ where $y_i = y_j$ and $y_i \neq y_k$, this formulation is an inversion of the training objective enforced by metric losses (maximize inter-class distances, minimize intra-class distances (Boudiaf et al., 2020)) for a given norm. Effectively, when $\mathbf{x}_i$ and $\mathbf{x}_j$ share classes $(y_i = y_j)$, the distance in the embedding space between these two points is maximized, while if $y_i \neq y_k$, the distance in the embedding space between $\mathbf{x}_i$ and $\mathbf{x}_k$ is minimized. The constrained optimization problem defined by $\rho(\cdot, \cdot, \cdot)$ can be solved using commonly approximation techniques for uncovering adversarial perturbations, such as FGSM (Goodfellow et al., 2015), C&W (Carlini & Wagner, 2017), PGD (Madry et al., 2018). To simplify notation in following equations, we adopt the notation of $\rho_{(i,j,k)}$ for $\rho((\mathbf{x}_i, y_i), (\mathbf{x}_j, y_j), (\mathbf{x}_k, y_k))$, in addition to $\rho_{(i,j)} = \rho_{(i,j,j)}$.

The robust training objective for the contrastive loss function be given by:

$$\underset{\theta \in \Theta}{\arg\min} \ l(\theta, (\mathbf{x}_1, y_1), (\mathbf{x}_2 + \gamma \rho_{(2,1)}, y_2)) \ . \tag{15}$$

Here, $\gamma \in \{0, 1\}$ is a discrete random variable for which $P(\gamma = 1) \in [0; 1]$ is a hyper-parameter that specifies the attack rate during training. For tuple-based losses perturbations are not performed on negative pairs, such that $P(\gamma = 1 \mid y_1 \neq y_2) = 0$. To reduce notation clutter, we use the notation of $P(\gamma = 1)$ interchangeably across losses, however, for tuple-based losses it refers to $P(\gamma = 1 \mid y_1 = y_2)$. Similarly, let the robust training objective for triplet-based metric losses be given by:

$$\underset{\theta \in \Theta}{\arg\min} \ l(\theta, (\mathbf{x}_1, y_1), (\mathbf{x}_2 + \gamma \rho_{(2,1)}, y_2), (\mathbf{x}_3, y_3)) \ . \tag{16}$$

These robust training objectives align the usual objective of adversarial training for deep neural networks, i.e. replace a sample by their "worst case" variant before normal training. For the sake of concreteness, we apply the objective to the $\ell_\infty$ norm, but the method is applicable to other norms commonly used for adversarial training.

## 4 EXPERIMENTS

Our experiments explore the following research questions.

**Q1.** How robust are *naturally* trained DML models towards established adversarial example attacks?

*Among commonly used datasets for visual similarity, we find that DML models, trained with established hyper-parameters, are vulnerable to adversarial examples, similar to non-DML models (Table 1). This forms our baseline for robustness.*

**Q2.** What is the accuracy of DML models when they are trained using our robust formulation?

*We find that DML models can be trained to become more robust towards a given threat model. For example, for a PGD attacker with 5 iterations and $\|\delta\|_\infty < 0.01$, we obtain an adversarial accuracy of 48.7 compared to the state-of-the-art natural accuracy baseline of 71.8 for the SOP dataset (Table 2).*

### 4.1 EXPERIMENTAL SETUP

Each experiment use parameter choices of Roth et al. (2020), unless otherwise specified, and perform evaluations on the commonly used datasets within DML. These choices are a reflecting of state-of-the-art performance for naturally-trained DML networks. We summarize some of these choices below, and emphasize deviations. For further we details on parameter choices, we refer the reader to Appendix A or the original work of Roth et al. (2020).

Experiments were executed on an NVIDIA Tesla V100 GPU (32GB RAM). Code for the experiments is available at: (anonymized repository) `https://github.com/starving-panda/adversarial-metric-learning`.

**Model & Optimizer.** Each experiment use a ResNet50 model (He et al., 2016) initialized with pre-trained ImageNet weights and frozen batch-norm layers. The last output layer is replaced with a fully-connected untrained embedding layer of size 128. ResNets are commonly used and preferable in DML due to their reduced amount of parameters (Musgrave et al., 2020; Roth et al., 2020), while having comparable performance to other more complex deep neural network (DNN) architectures. For training, the ADAM (Kingma & Ba, 2015) optimizer with a learning rate[3] of $10^{-6}$, weight decay of $4 \cdot 10^{-4}$ is used. For contrastive loss $\alpha = 1.0$, while $\alpha = 0.2$ for triplet loss.

**Datasets.** We use datasets common for metric learning: (1) CUB200-2011 (Welinder et al., 2010) containing 11788 images of birds across 200 species; (2) CARS196 (Krause et al., 2013) containing 16185 images of cars across 196 different models; (3) SOP (Song et al., 2016) containing 120053 images of 22634 different online products. Each dataset is divided into a training and testing set of approximately the same size, by respective selecting the first half of classes for the training set, while having the remaining classes be the testing set. This split reflect a zero-shot learning scenarios, which is a common application of DML. Similar to Roth et al. (2020), we train CUB200-2011 and CARS196 for 150 epochs and SOP for 100 epochs due to its volume.

**Adversarial Perturbations.** The proposed attack algorithm, Algorithm 1, and $\rho(\cdot, \cdot, \cdot)$ depends on solving an $\arg\max$ expression. To solve this formulation, we use PGD (Madry et al., 2018), which is an established attack method for creating adversarial perturbations. Attacks are run for five iterations, this was chosen to keep the efficient (in terms of runtime), as training is DML models is considered an expensive procedure (Roth et al., 2020), and because the performance of the attack is known to saturate with high iterations counts (Wong et al., 2020). Step sizes are fixed to $2(\frac{\epsilon}{j})$, where $\epsilon$ specifies the domain of valid perturbations under $\ell_\infty$, thus any perturbation $\delta$ satisfy $\|\delta\|_\infty \leq \epsilon$. This step size was chosen to keep the step size small, while ensuring the any point within the $\epsilon$-ball is reachable, regardless of initialization (Wong et al., 2020). Alternative attack methods, and their effectiveness towards naturally-trained DML models, can be seen in Appendix D.

**Evaluation Metrics.** We use two DML-specific evaluation metrics proposed by Musgrave et al. (2020), namely: Recall at one (R@1) and Mean Average Precision at R (mAP@R). R@1 is the accuracy of using the embeddings that the model produces for inferring the output class label using the class of the nearest neighbor anchor. Given a test set $S = \{(\mathbf{x}_1, y_1), \cdots, (\mathbf{x}_n, y_n)\}$, R@1 is given by:

$$\text{R@1} = \frac{1}{n} \sum_{i=1}^{n} \text{Prec}(i,1) \ , \text{ where } \text{Prec}(i,k) = \frac{1}{k} \sum_{j \in I(k)} 1_{y_i = y_j} \tag{17}$$

$$I(k) = \underset{\substack{|K|=k \\ i \notin K}}{\arg\min} \sum_{j \in K} d_\theta(\mathbf{x}_i, \mathbf{x}_j) \tag{18}$$

mAP@R (Musgrave et al., 2020) measures a given model's ability to rank among classes in the embedding space; we adopted this metric for the reasons covered by Musgrave et al. (2020). It is defined as:

$$\text{mAP@R} = \frac{1}{n} \sum_{i=1}^{n} \frac{1}{R(i)} \sum_{k=1}^{R(i)} \text{Prec}(i,k) \ , \text{ where } R(i) = \sum_{\substack{(x_j, y_j) \in S \\ i \neq j}} 1_{y_i = y_j} \tag{19}$$

### 4.2 EXPERIMENTAL RESULTS

First we seek to establish a baseline of robustness against adversarial perturbations for naturally-trained DML models. We obtain this baseline by applying Algorithm 1 for three common values

---

[3]This learning rate differ from the one stated by Roth et al. (2020) in their publication, $10^{-5}$, but reflects the actual learning rate used throughout their experiments. See the field "lr" within experiment configuration: https://bit.ly/3a4FyHP.

| | $\ell_\infty(\epsilon = \downarrow)$ | CUB200-2011 | | CARS196 | | SOP | |
|---|---|---|---|---|---|---|---|
| | | R@1 | mAP@R | R@1 | mAP@R | R@1 | mAP@R |
| Contrastive | *Unperturbed* | *59.1 ± 0.0* | *21.0 ± 0.0* | *74.0 ± 0.0* | *20.9 ± 0.0* | *71.8 ± 0.0* | *44.7 ± 0.0* |
| | 0.01 | 2.1 ± 0.1 | 2.6 ± 0.0 | 0.4 ± 0.0 | 1.5 ± 0.0 | 1.3 ± 0.0 | 1.7 ± 0.0 |
| | 0.05 | 0.1 ± 0.0 | 2.2 ± 0.0 | 0.2 ± 0.0 | 1.4 ± 0.0 | 0.0 ± 0.0 | 1.1 ± 0.0 |
| | 0.10 | 0.0 ± 0.0 | 2.2 ± 0.0 | 0.2 ± 0.0 | 1.4 ± 0.0 | 0.0 ± 0.0 | 1.1 ± 0.0 |
| Triplet | *Unperturbed* | *59.3 ± 0.0* | *21.7 ± 0.0* | *74.0 ± 0.0* | *21.4 ± 0.0* | *69.6 ± 0.0* | *42.1 ± 0.0* |
| | 0.01 | 2.5 ± 0.1 | 3.0 ± 0.0 | 0.3 ± 0.0 | 1.6 ± 0.0 | 0.5 ± 0.0 | 1.4 ± 0.0 |
| | 0.05 | 0.0 ± 0.0 | 2.5 ± 0.0 | 0.1 ± 0.0 | 1.5 ± 0.0 | 0.0 ± 0.0 | 1.2 ± 0.0 |
| | 0.10 | 0.0 ± 0.0 | 2.5 ± 0.0 | 0.1 ± 0.0 | 1.5 ± 0.0 | 0.0 ± 0.0 | 1.2 ± 0.0 |

Table 1: Performance of naturally-trained DML models against adversarial perturbations generated using Algorithm 1 with PGD across various $\epsilon$ for $\ell_\infty$. Results are an average of five random seeds. Recall that, R@1 reflects a model's inference accuracy, while mAP@R reflects its ability to rank similar entities. Naturally-trained DML models are not robust to the generated adversarial perturbations.

| $\downarrow$Loss | $\ell_\infty(\epsilon = 0.01)$ | CUB200-2011 | | CARS196 | | SOP | |
|---|---|---|---|---|---|---|---|
| | | R@1 | mAP@R | R@1 | mAP@R | R@1 | mAP@R |
| C | Natural | 2.1 ± 0.1 | 2.6 ± 0.0 | 0.4 ± 0.0 | 1.5 ± 0.0 | 1.3 ± 0.0 | 1.7 ± 0.0 |
| | **Robust** | **18.7 ± 0.3** | **7.5 ± 0.0** | 41.4 ± 0.0 | **11.5 ± 0.0** | **48.7 ± 0.0** | **28.3 ± 0.0** |
| T | Natural | 2.5 ± 0.1 | 3.0 ± 0.0 | 0.3 ± 0.0 | 1.6 ± 0.0 | 0.5 ± 0.0 | 1.4 ± 0.0 |
| | **Robust** | **23.0 ± 0.2** | **8.6 ± 0.0** | **40.7 ± 0.0** | **11.9 ± 0.0** | **38.0 ± 0.1** | **21.1 ± 0.0** |

Table 2: Performance of DML models trained using the proposed adversarial training objective (using PGD) compared to naturally-trained DML for adversarial perturbations within $\ell_\infty(\epsilon = 0.01)$. Losses are denoted by C (contrastive) and T (triplet). The robustly trained model attain both higher inference accuracy (R@1) and improved ability to rank similar entities (mAP@R) than the naturally-trained baseline model. Thereby, the proposed robust training objective improves the robustness towards adversarial perturbations.

of $\epsilon \in \{0.01, 0.05, 0.1\}$ used for computer vision[4], under the $\ell_\infty$ norm, such that a perturbation $\delta$ satisfy $\|\delta\|_\infty \leq \epsilon$. PGD with five iterations is used to approximate a solution to the $\arg\max$. Five iterations was chosen as (1) it saturates the maximum attack effect because R@1 $\approx 0$, and (2) in this situation, adding more iterations leads to stagnation (Wong et al., 2020). Table 1 summarizes the findings, for which reported performance is an averaged over five distinct random seeds for the attacks algorithm.

We observe that naturally-trained DML models achieve very low robustness to our proposed attack, experiencing a drop in both R@1 (inference) and mAP@R (ability to rank) by several orders of magnitude. Thus, to answer Q1, we conclude that DML models are not inherently robust to established attacks, suggesting that prior studies which report higher robustness of naturally-trained DML models is a potential sign of ineffective attacks instead (Abdelnabi et al., 2020).

Next, we examine the effectiveness of our robust training objective (Section 3.5) for creating robust DML models. We set a threat model $\ell_\infty(\epsilon = 0.01)$ and optimize a min-max objective while re-using hyper-parameter choices from non-robust training for five attack rates $P(\gamma = 1) \in \{0.1, 0.25, 0.5, 0.75, 1.0\}$. For the inner-maximization, we use PGD with a similar configuration to the attack setting. Table 2 summarizes the results for the best performing robust model, for which the performance is reported as an average of five attacks with distinct random seeds. Performance of the trained robust models on natural (unperturbed) input largely remains unaffected, for more details on this, we refer the reader to Appendix C.

We observe that our robust formulation does improve model robustness against the specified PGD-based attack across any combination of loss and dataset. Consistently, for all of the considered

---

[4]https://www.robust-ml.org/defenses/

attack rates, the robustness increased, more details on this matter can be found in Appendix C. An example of inference across training objectives can be seen in Figure 1. For more examples we refer the reader to the online gallery [5]. We also observed that the gained robustness towards the PGD-based also transfer to other established attack methods, such as C&W and FGSM, see Appendix B for related results. More details on the effects of the robust training objective on the embedding space, is provided by a small experiment using a high-dimensional synthetic data set in Appendix E, highlighting the embeddings of a robust model remain more stationary when input is adversarially perturbed.

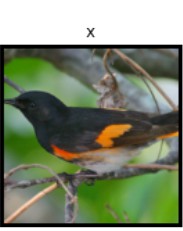
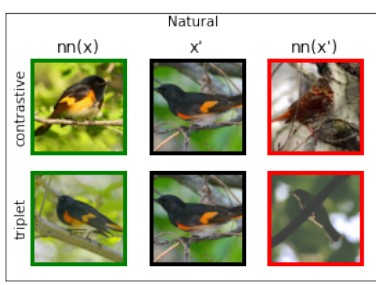
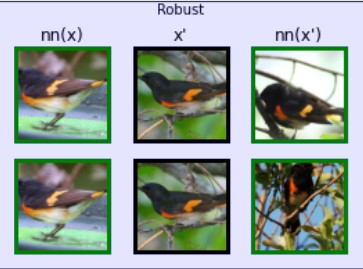

Figure 1: Example of inference of a naturally-trained DML model and robustly trained variant on the CUB200-2011 dataset. Each model infer the class of the natural data point $\mathbf{x}$, and its perturbed $\mathbf{x}'$ counterpart, using the class of the nearest anchor $nn(\cdot)$. Green and red borders indicate correct-(same class) and incorrect inference, respectively. Both models infer the natural input correctly, however, the naturally-trained DML fails to infer the adversarial perturbed input correctly.

## 5 DISCUSSION

The proposed robust training objective relies on perturbing positive data points within tuples and triplets, for reasons addressed in Section 3.4. Experimentation of alternative perturbation targets, in Appendix F, demonstrate that the choice of perturbation target (positive) out perform alternatives (anchor, positive) in five out of six cases. For one particular case (contrastive loss on CUB200-2011), perturbing the anchor data points achieves better performance (R@1 = 20.9) than the proposed method (R@1 = 18.4). We speculate that the inability of other perturbation targets are associated with known instabilities of metric losses (Wu et al., 2017). Namely, that large distances between anchors and negatives cause gradient updates during training to be noisy across batches. Stability is typically achieved by applying a sampling process that account for this phenomenon, and thus select a certain distribution of tuples or triplets. The proposed robust training objective induce noise (high loss) post-sampling, potentially causing instabilities during training. We hypothesis that this fact could be related to the observed drop in performance for certain training scenarios with high attack rates, $P(\gamma = 1) = 1$, as seen in Appendix C. Despite the proposed robust training objective already demonstrating promising results, we encourage future research to explore the inherent dilemma between robust optimization and instabilities in DML further.

## 6 CONCLUSION

Deep Metric Learning (DML) creates feature embedding spaces where similar input points are geometrically close to each other, while dissimilar points are far apart. However, the underlying DNNs are vulnerable to adversarial inputs, thus making the DML models themselves vulnerable. We demonstrate that naturally-trained DML models are vulnerable to strong attackers, similar to other types of deep learning models. To create robust DML models, we contribute a robust training objective that can account for the *dependence* of metric losses — the phenomenon that the loss at any point depends on the other items in the mini-batch and the sampling process that was used to derive the mini-batch. Our robust training formulation yields robust DML models that can withstand powerful PGD attackers without severely degrading their performance on natural inputs.

---

[5](anonymized gallery) `https://starving-panda.github.io/sample-gallery/`

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

## A  TRAINING PARAMETERS (EXPANDED)

This section expands upon details and hyper-parameters used throughout the training of the respective DML models.

**Batches & Sampling.**    The training process use a mini-batch size of 112 data points. Each mini-batch is sampled such that it contains a fixed amount of classes per class (Roth et al., 2020), for which this fixed amount is specified is two samples per classes. Following this, sets of tuples and triplets (depending on loss used for training) are derived from the mini-batch. The triplet-set's size is the same as the mini-batch, for which each data point is used as an anchor. The tuple-set's size is double that of the mini-batch, thus balancing out the number of data points being compared relative to the triplet-set. Furthermore, each data point within the tuple-set is used in a positive and negative pair. DML models are prone to collapse during training, a phenomenon caused by too many distant negative samples in the tuple- or triplet-sets, causing the model to map any data point to the same point in the embedding space. As a measure to reduce the probability of facing this phenomenon, distance-weighted sampling is used for sampling negatives (Wu et al., 2017).

**Data Augmentation.**    We augment the dataset using the following operations for each input image: (1) random cropping to an image patch of size 60-100% of the original image area; (2) scaling; (3) normalization of pixel intensities. One difference is that our patch sizes differ from Roth et al. (2020) that employs patches of size 8-100% of original area. We change this parameter because recent work suggests that computer vision models can be biased by backgrounds and textures during during (Xiao et al., 2020). To combat this, we use cropping and scaling values based on Szegedy et al. (2015).

## B  EXPERIMENT: UNIVERSALITY OF ROBUSTNESS

The robust DML models trained using the proposed adversarial training algorithm (covered in Section 3.5) approximate the solution to the $\arg\max$ in $\rho((\mathbf{x}_i, y_i), (\mathbf{x}_j, y_j), (\mathbf{x}_k, y_k))$ (defined in Equation 14) using PGD (Madry et al., 2018), and have demonstrated improved robustness towards attacks using on PGD (see Table 2). However, as a measure to test if the increased robustness also apply to attacks relying on other attack methods, such as FGSM and C&W, we conduct a robustness evaluation using Algorithm 1 with these two attack methods for $\ell_\infty(\epsilon = 0.01)$. For FGSM, we use $\alpha = \epsilon$, to maximize the perturbation, and thereby attack strength, under the given norm. Despite C&W being originally designed for the $\ell_2$ norm, we sought out to adopt an implementation created by the author [6] for the $\ell_\infty$ norm. However, it must be noted that the author now discourage the use of the attack method in favor of PGD for $\ell_\infty$ [7]. For C&W, we had to limit the attack to 50 iterations as a measure to combat extreme runtime, that were infeasible to run on our available resources. Thus the applied attack might not be able capture the full potential of the attack method. Results of these experiments can be seen in Table 3 and Table 4. It can be seen that, despite having used PGD for the adversarial training, the improved robustness is also useful for perturbations of other attack methods.

## C  EXPERIMENT DETAILS: ATTACK RATES

This section expands upon the training process of the proposed robust formulation, particularly the impact of attack rate $P(\gamma = 1)$, and the following performance of the DML models. In Table 5, the performance of models against adversarial perturbations across various attack rates can be seen. These results demonstrate that robustness of the models being trained increases across all of the covered attack rates $P(\gamma = 1) \in \{0.1, 0.25, 0.5, 0.75, 1.0\}$. Additionally, there seems to be an association between higher attack rates yielding higher robustness. However, this relationship is not monotonic, and across data sets and losses optimal choices for attack rate vary. In Table 6, the

---

[6]C&W for $\ell_\infty$, TensorFlow Implementation: `https://github.com/carlini/nn_robust_attacks/blob/master/li_attack.py`

[7]Nicholas Carlini on C&W for $\ell_\infty$: `https://github.com/tensorflow/cleverhans/issues/978#issuecomment-464594668`

| Loss↓ FGSM | | CUB200-2011 | | CARS196 | | SOP | |
|---|---|---|---|---|---|---|---|
| | | R@1 | mAP@R | R@1 | mAP@R | R@1 | mAP@R |
| C | Natural | 19.7 | 7.2 | 14.6 | 3.8 | 18.4 | 10.8 |
| | **Robust** | **24.6** | **9.9** | **44.6** | **12.4** | **50.3** | **29.2** |
| T | Natural | 18.9 | 7.7 | 15.5 | 4.1 | 14.5 | 8.5 |
| | **Robust** | **27.4** | **11.0** | **44.1** | **13.0** | **43.2** | **24.2** |

Table 3: Performance of DML models trained using the proposed adversarial training objective (using PGD) compared to naturally-trained DML for adversarial perturbations within $\ell_\infty (\epsilon = 0.01)$ discovered using FGSM. Losses are denoted by C (contrastive) and T (triplet). The robustly trained model attain both higher inference accuracy (R@1) and improved ability to rank similar entities (mAP@R) than the naturally-trained baseline model. Thereby, the proposed robust training objective improves the robustness towards adversarial perturbations.

| Loss↓ C&W | | CUB200-2011 | | CARS196 | | SOP | |
|---|---|---|---|---|---|---|---|
| | | R@1 | mAP@R | R@1 | mAP@R | R@1 | mAP@R |
| C | Natural | 15.1 | 6.0 | 5.6 | 2.3 | 23.1 | 13.9 |
| | **Robust** | **37.4** | **14.7** | **58.6** | **15.3** | **57.2** | **32.6** |
| T | Natural | 22.9 | 10.0 | 13.4 | 3.6 | 10.0 | 6.1 |
| | **Robust** | **39.9** | **16.2** | **58.9** | **16.0** | **55.1** | **32.2** |

Table 4: Performance of DML models trained using the proposed adversarial training objective (using PGD) compared to naturally-trained DML for adversarial perturbations within $\ell_\infty (\epsilon = 0.01)$ discovered using C&W. Losses are denoted by C (contrastive) and T (triplet). The robustly trained model attain both higher inference accuracy (R@1) and improved ability to rank similar entities (mAP@R) than the naturally-trained baseline model. Thereby, the proposed robust training objective improves the robustness towards adversarial perturbations.

performance for the robust (and a naturally-trained) models against benign (unperturbed) input can be seen. Generally, it can be seen that the robust formulation has minor impact on performance on benign input, as each model's evaluation metrics are close to performance of the naturally-trained baseline. Interestingly, for contrastive loss on CUB200-2011 and CARS196 ($P(\gamma = 1) = 0.25$ and $P(\gamma = 1) = 0.1$), the robust training objective have enable the benign performance to exceed the naturally-trained baseline. This could suggest that a low frequency of adversarial perturbations could potentially improve the training process of non-robust DML. We deem more details on this to out of scope for our work, but see it as an promising direction for future research to explore more thoroughly.

## D  EXPERIMENT: NATURAL ROBUSTNESS (ALTERNATIVES)

This section covers the robustness for naturally-trained DML models for two other established attack methods, FGSM (Goodfellow et al., 2015) and C&W (Carlini & Wagner, 2017). Using the Algorithm 1 with these methods, it can be seen in Table 7 that they also lower the performance for $\ell_\infty (\epsilon = 0.01)$. Hyper-parameters for C&W are covered in Appendix B, particularly iterations had to remain low to feasible runtime under the available resources. Thereby, tuning parameters to a greater extend could yield a more powerful attack and thus lower robustness.

## E  EXPERIMENT: SYNTHETIC DATA

As a measure to establish more clarity into the effects of induced by the robust training objective (covered in Section 3), we seek to construct an experiment with two high-dimensional and well-defined data distributions being mapped to low-dimensional embedding space. We define the dimensionality of the high-dimensional input space as $k = 3 \times 224 \times 224$, in order to share the dimensionality of the other experiments using real-world datasets. We construct the dataset with

| $P(\gamma=1)\downarrow$ | | CUB200-2011 | | CARS196 | | SOP | |
|---|---|---|---|---|---|---|---|
| | | R@1 | mAP@R | R@1 | mAP@R | R@1 | mAP@R |
| Contrastive | *Naturally-trained* | 2.1 | 2.6 | 0.4 | 1.5 | 1.3 | 1.7 |
| | 0.1 | 8.0 | 4.0 | 8.5 | 2.7 | 21.3 | 12.2 |
| | 0.25 | 17.9 | 6.8 | 27.1 | 7.2 | 39.8 | 23.7 |
| | 0.5 | **18.4** | **7.5** | 35.7 | 10.2 | 44.9 | 26.2 |
| | 0.75 | **18.4** | 7.3 | 39.5 | 11.4 | **48.7** | **28.3** |
| | 1.0 | 15.4 | 6.5 | **41.4** | **11.5** | 44.0 | 24.9 |
| Triplet | *Naturally-trained* | 2.5 | 3.0 | 0.3 | 1.6 | 0.5 | 1.4 |
| | 0.1 | 22.3 | **8.8** | 29.7 | 8.0 | 26.8 | 15.2 |
| | 0.25 | 22.1 | **8.8** | 32.7 | 9.2 | 32.2 | 18.4 |
| | 0.5 | **23.1** | 8.6 | 36.7 | 10.7 | **38.1** | **21.1** |
| | 0.75 | 20.4 | 7.7 | 39.7 | 11.6 | **38.1** | 20.9 |
| | 1.0 | 8.7 | 3.7 | **40.6** | **11.9** | 28.8 | 15.4 |

Table 5: Performance of robust DML models on adversarial input for various specified (training) attack rates $P(\gamma=1)$. These models were trained using the proposed adversarial training algorithm (covered in Section 3.5) with PGD for $\ell_\infty(\epsilon=0.01)$. Evaluations on conducted on adversarial input generated using Algorithm 1. Naturally-trained marks the performance of DML models using traditional non-robust training objectives. **Bold** marks best performance for dataset, metric, loss combinations. Robust models reach higher inference accuracy (R@1) and better ability to rank similar entities (mAP@R) on adversarial input than naturally-trained DML models. Higher attack rates are often associated with higher robustness.

| $P(\gamma=1)\downarrow$ | | CUB200-2011 | | CARS196 | | SOP | |
|---|---|---|---|---|---|---|---|
| | | R@1 | mAP@R | R@1 | mAP@R | R@1 | mAP@R |
| Contrastive | *Naturally-trained* | *59.1* | *21.0* | *74.0* | ***20.9*** | ***71.8*** | ***44.7*** |
| | 0.1 | 54.5 | 17.2 | **74.4** | **20.9** | 69.7 | 42.2 |
| | 0.25 | **60.1** | **21.9** | 74.0 | 20.3 | 66.7 | 39.2 |
| | 0.5 | 52.1 | 15.8 | 71.8 | 18.3 | 65.1 | 37.3 |
| | 0.75 | 50.7 | 14.8 | 70.1 | 17.3 | 62.0 | 34.4 |
| | 1.0 | 48.3 | 13.2 | 67.7 | 15.9 | 59.7 | 32.1 |
| Triplet | *Naturally-trained* | ***59.3*** | ***21.7*** | ***74.0*** | ***21.4*** | ***69.6*** | ***42.1*** |
| | 0.1 | 51.7 | 16.9 | 72.2 | 19.2 | 67.0 | 39.1 |
| | 0.25 | 52.2 | 16.8 | 70.9 | 18.4 | 65.2 | 37.2 |
| | 0.5 | 53.8 | 17.4 | 71.7 | 18.9 | 64.9 | 36.6 |
| | 0.75 | 53.0 | 16.5 | 70.3 | 18.1 | 63.5 | 35.3 |
| | 1.0 | 47.5 | 13.0 | 70.2 | 17.3 | 56.7 | 29.0 |

Table 6: Performance of robust DML models on benign input for various specified (training) attack rates $P(\gamma=1)$. These models were trained using the proposed adversarial training algorithm (covered in Section 3.5) with PGD for $\ell_\infty(\epsilon=0.01)$. Evaluations on benign data. Naturally-trained marks the performance of DML models using traditional non-robust training objectives. **Bold** marks best performance for dataset, metric, loss combinations. Higher attack rates are often associated with lower performance on benign input. For contrastive loss

| $\ell_\infty(\epsilon = 0.01)$ | | CUB200-2011 | | CARS196 | | SOP | |
|---|---|---|---|---|---|---|---|
| Loss ↓ | | R@1 | mAP@R | R@1 | mAP@R | R@1 | mAP@R |
| C | *Benign* | 59.1 | 21.0 | 74.0 | 20.9 | 71.8 | 44.7 |
| | FGSM | 19.7 | 7.2 | 14.6 | 3.8 | 18.4 | 10.8 |
| | C&W | 15.1 | 6.0 | 5.6 | 2.3 | 23.1 | 13.9 |
| T | *Benign* | 59.3 | 21.7 | 74.0 | 21.4 | 69.6 | 42.1 |
| | FGSM | 18.9 | 7.7 | 15.5 | 4.1 | 14.5 | 8.5 |
| | C&W | 22.9 | 10.0 | 13.4 | 3.6 | 10.0 | 6.1 |

Table 7: Performance of naturally-trained DML models against adversarial examples generated using Algorithm 1 with two alternative attack methods: FGSM and C&W. Losses are denoted by C (contrastive) and T (triplet). Recall that, R@1 reflects a model's inference accuracy, while mAP@R reflects its ability to rank similar entities.

two classes, "a" and "b", for which each has an associated independent $k$-dimensional Gaussian distribution. These distributions are $\mathcal{N}_k(\mu_a, \Sigma)$ and $\mathcal{N}_k(\mu_b, \Sigma)$, for class "a" and "b" respectively, where:

$$\mu_a = (0.25 \quad \cdots \quad 0.25) \in \mathbb{R}^k, \quad \mu_b = (0.75 \quad \cdots \quad 0.75) \in \mathbb{R}^k, \text{ and } \Sigma = \sigma^2 \cdot I_k . \qquad (20)$$

Here, $I_k \in \mathbb{R}^{k \times k}$ is the identity matrix of size $k$ and $\sigma = 0.025$. Following this, we draw a dataset $D_\mathcal{N}$ with a fixed amount of data points for each class and train a deep metric model $f_\theta \colon \mathbb{R}^k \to \mathbb{R}^2$ parameterized by $\theta$. We choose to have the embedding space be two-dimensional as a measure to enable visualizations of the learned embedding space. Using $D_\mathcal{N}$, we train two variants of $f_\theta$, one using the natural (non-robust) training objective and another using the proposed robust training objective (for $\ell_\infty(\epsilon = 0.01)$). Each model use contrastive loss, is trained across 25 epochs on 508 train data points, while being evaluated on 516 test data points. Following, we perform a test-time attack using Algorithm 1 with PGD ($\epsilon = 0.01$). Differences of the learned embedding spaces, and the influence of the adversarial perturbations, can be seen in Figure E.

Generally, it can be seen that embeddings of the robust model remain more stable (in terms of position) in the embedding space, when compared to naturally-trained model. Both models have R@1 = 100.0 for benign (unperturbed) data, while the naturally-trained model have R@1 = 4.6 for adversarial data points, while the robust model attains R@1 = 100.0 for those data points.

## F  EXPERIMENT: PERTURBATION TARGET

The proposed adversarial training techniques for DML apply perturbations to positive data points. This choice was made to establish alignment with the adversarial behavior covered in Section 3.4. However, the "adversarial loss" $\rho(\cdot, \cdot, \cdot)$ can capture violations across various data point types (anchor, positive, negative). To assess the impact of using alternative perturbation targets (negative, anchor) for adversarial training, we performed an experiment with $P(\gamma = 1) = 0.5$ (unless otherwise stated) across these alternative perturbation targets. Losses and attack rates used during adversarial training, for the other perturbation targets, are specified below.

Negative perturbations for contrastive loss:

$$l(\theta, (\mathbf{x}_1, y_1), (\mathbf{x}_2 + \gamma\rho_{(2,1)}, y_2)) ,$$

$$P(\gamma = 1 \mid y_1 = y_2) = 0 , \text{ and } P(\gamma = 1 \mid y_1 \neq y_2) = 0.5 .$$

Anchor perturbations for contrastive loss:

$$l(\theta, (\mathbf{x}_1 + \gamma\rho_{(1,2)}, y_1), (\mathbf{x}_2, y_2)) ,$$

$$P(\gamma = 1 \mid y_1 = y_2) = P(\gamma = 1 \mid y_1 \neq y_2) = 0.5 .$$

Negative perturbations for triplet loss:

$$l(\theta, (\mathbf{x}_1, y_1), (\mathbf{x}_2, y_2), (\mathbf{x}_3 + \gamma\rho_{(3,1)}, y_3))$$

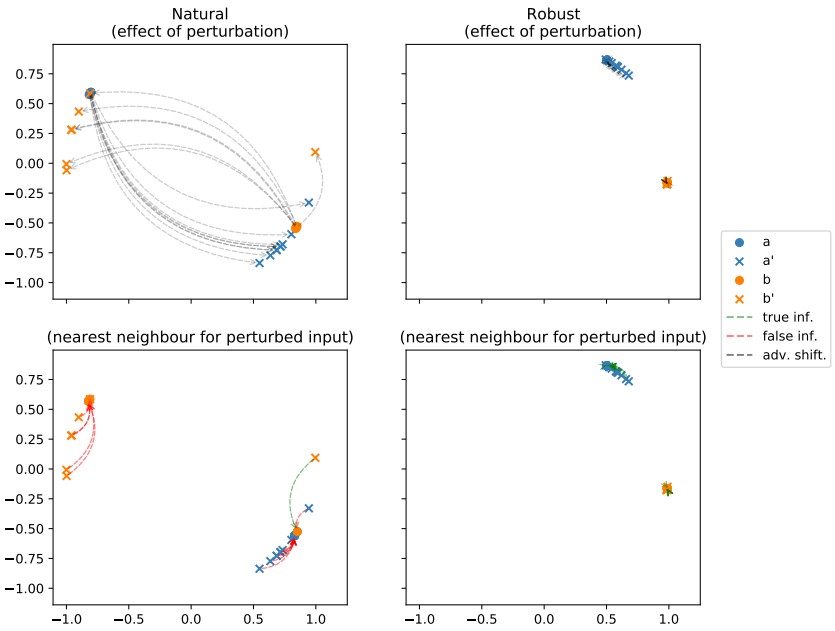

Figure 2: Effects of adversarial perturbations in the embedding space (embedding shift, inference) for 16 randomly sampled data points. Circles mark embeddings of unperturbed data points, while crosses are adversarial perturbations for the respective data points. (first row) Grey lines connect the unperturbed data points to their perturbed counterpart, to highlight the shift in the embedding space. (second row) Green and red lines connect the embedding of the respective adversarial data point to its nearest (and unperturbed) neighbor. If the line is green, this neighbor is of the same class (correct) while red is the opposite (wrong). It can be seen that embeddings of the robustly-trained model shifts much less, when faced with adversarial perturbed input, and are thus more robust.

Anchor perturbations for triplet loss:

$$l(\theta, (\mathbf{x}_1 + \gamma\rho_{(1,2,3)}, y_1), (\mathbf{x}_2, y_2), (\mathbf{x}_3, y_3))$$

Training processes for contrastive- and triplet loss can be seen in Figure 3 and Figure 4, respectively. Across each combination of loss and dataset, the proposed adversarial training (positive perturbation) yields the most robustness for five out six experiments, with the exception of CUB200-2011 for contrastive loss. Additionally, it reaches several orders of magnitudes higher performance on the CARS196 and SOP datasets. We speculate that this inability for the alternative perturbation targets to reach similar performance can be linked to alterations on distance between anchors and negatives during training. Having large distance between anchor data points and negative data points are known to cause instabilities during training, causing models to reach a local minima (Wu et al., 2017).

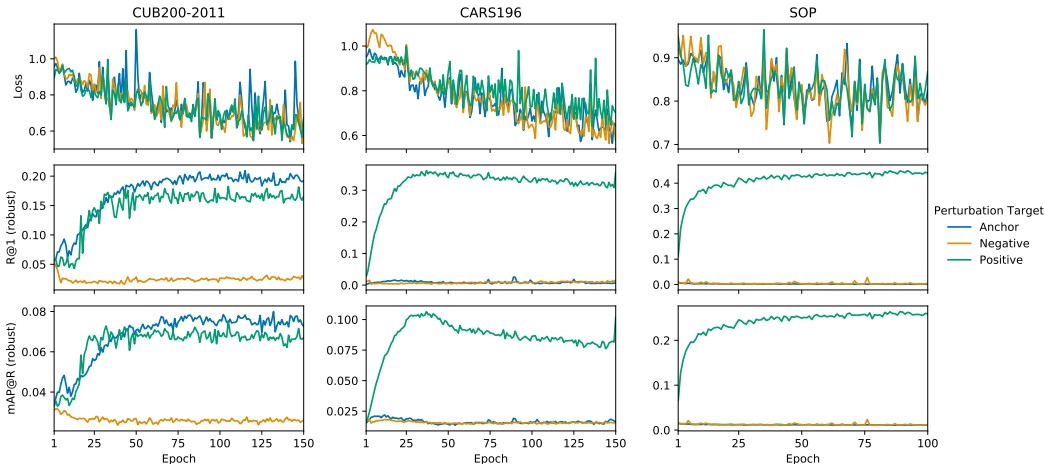

Figure 3: Metrics (Loss, R@1, and mAP@R) for contrastive loss training procedure across perturbation targets.

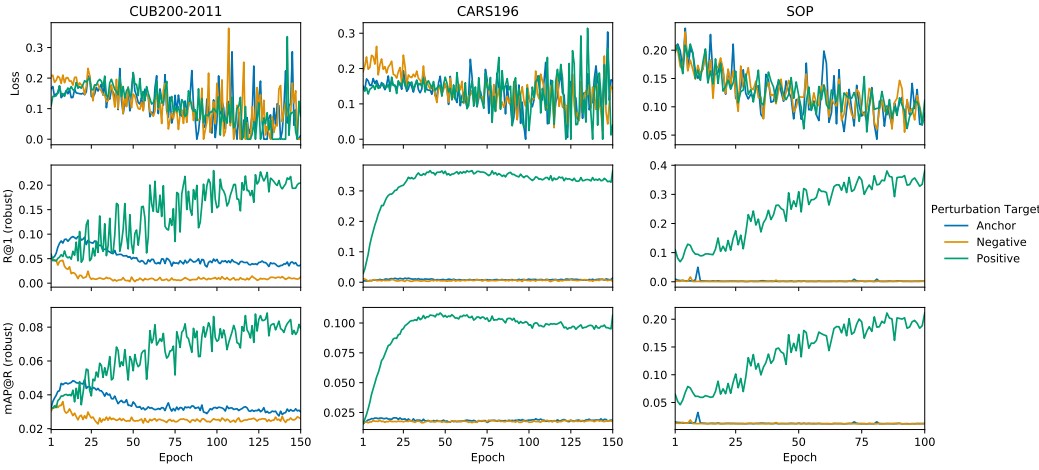

Figure 4: Metrics (Loss, R@1, and mAP@R) for triplet loss training procedure across perturbation targets.

