# OpenReview forum: "Adversarial Deep Metric Learning"
_ICLR.cc/2021/Conference — Reject_

### Official Review · AnonReviewer1 · 2020-10-28
**Interesting idea, promising results, though it seems a bit rushed**

**Rating:** 6
**Confidence:** 4

**Review:**


The authors propose a novel robust training approach for deep metric learning (DML), accounting for the dependencies of metric losses within mini-batches. The proposed approach is evaluated on several popular metric learning datasets, demonstrating that the method works as intended, and achieves a certain level of robustness, unlike the baseline non-robust model which achieves a very poor performance when exposed to an adversarial attack.

Comments:

In the opening paragraphs of the paper, the authors state ‘Our key insight is that during an inference-time attack, the positive point in the triplet will be modified by the attacker, and thus, during training, we must perturb positive points in the triplets, instead of anchors or negative points.’, yet towards the end (and despite having motivated it in between), they show that the results on this are inconclusive and that it is not clear that perturbing only the positive points in the triplets is the right / best thing to do. Unless I am misunderstanding something, this feels out of sync - either it is a key insight, or inconclusive? It can’t be both.

Opening paragraphs / under Q1 and Q2 in Section 4 (Experiments) - this feels both redundant (both were mentioned before under Contributions) and also out of place, as it mentions results before even introducing the experimental setup.

The authors evaluate their proposed robustness approach under a projected gradient descent (PGD) attack. While this is certainly sufficient to establish that the method works and that it provides some level of robustness to adversarial attacks, it feels really limiting to only assess a single attack type, of various options that are available - as it would have been potentially valuable to establish the degree of provided robustness under these different cases. As is, it is unclear whether the proposed approach will be universally helpful, or merely helpful with a particular attack type. While there is no reason to believe in the latter, the former hasn’t been substantiated nor argumented.

Apart from the general metrics shown in the tables, there isn’t much additional analysis that would aim to reveal whether there were any patterns in these datasets on where the method worked vs didn’t. For example, was the performance uniform across (pairs of) classes? If not, why? What about contrasting class pairs that are more/less similar? Or are there issues with rare classes?

Tables 1, 2 and 3 should include confidence intervals.

The authors use the phrase significant to qualify differences in several parts of the paper, yet there is no mention of which statistical test has been used to claim statistical significance of the differences? The authors should conduct proper statistical testing and highlight the exact test in the text.

There is no discussion of the limitations of the current approach / areas for potential improvement and future work in the main paper - instead, some open questions are mentioned in the Appendix. It would be good to include key discussion points in the main body of the paper.

Nit: Page 3, when giving lb(A,z) = c_k(A,z) - what if there is more than a single index returned by the argmin - what if there is a tie? The authors should specify if they are doing random tie breaking or taking the majority label (if a multi-way tie)

Nit: Figure 4, please update with the finalized results.

---

> ### Author Response · Authors · 2020-11-19
> **Initial response**
>
> We are thankful for the comments and pleased to hear you found our work novel.
>
> *Using the $P_n$ notation for referencing paragraphs, where $n$ is the paragraph being referenced.*
>
> ($P_3$)  Our choice of PGD were merely to utilize an efficient method of obtaining effective adversarial perturbations by solving  argmax in the provided formulation. When you mention that `unclear whether the proposed approach will be universally helpful, or merely helpful with a particular attack type. `, are you referring helpfulness in the sense of  (1) can robustness [to a certain extend] be achieved using other attack methods (e.g. FGSM, C&W), or (2) is the achieved robustness universal towards other perturbations obtained from other established attack algorithms (e.g. FGSM, C&W)?
>
> ($P_4$)  We find your suggested analysis proposal a very interesting idea. Our incentive was to provide the field of DML (using metric losses) with methods for (a) obtaining adversarial perturbations and (b) enhancing robustness of models undergoing training. However, we fear that the proposed idea would shift the focus towards an analysis of the respective data distributions of the real-world data sets, rather than seeing failure modes of the proposed methods for obtaining robustness.
>
> ($P_6$) Thanks for raising this concern. We will provide a more precise wording in the respective scenarios.

---

> ### Author Response · Authors · 2020-11-24
> **Summary of changes**
>
> Throughout our revision, we have improved upon the following related to your feedback:
> - We have address the stated contradiction about the key insight, and clarified our stance on this throughout the paper and address it in the discussion (now main body). Importantly, as we have updated our implementation (and rerun experiments) the experiments now yield different results.
> - To address the question about the *universality* of the gained robustness, we have provided additional evaluations using other attack methods (C&W, FGSM) and include the performance in Appendix (see Table 3 and Table 4).
> - To provide more clarity into the effects of the robust training objective, we have included  an experiment on a high-dimensional synthetic data (embedding size = 2, for visualization purposes), highlighting how the robust training objective affects the learned embedding space. See Appendix E. Additionally, we have also added a section on the ability for the robust models to remain robust towards other attack methods (see Appendix B).
> - We have updated Table 1 with confidence intervals. Table 2 have been completely revised, and now include confidence intervals.
> Previous Table 2 have been moved to Table 5. With the resources we have available, it have unfortunately been infeasible to add confidence intervals for this, it requires approximately 150 hours of effective GPU runtime.
> - We have revised the discussion and added it to the main body of the paper.
> - Provided an [online gallery](https://starving-panda.github.io/sample-gallery/) for examples of nearest neighbor inference.
>
> *We have revised our implementation, thus experiments has been rerun and numbers are updated throughout the paper.*

---

### Official Review · AnonReviewer4 · 2020-10-28
**First results for robust deep metric learning**

**Rating:** 6
**Confidence:** 4

**Review:**

Summary:

This paper analyzes the robustness to adversarial attacks of deep metric learning (DML) models for image similarity. A robust training framework is proposed. Experiments are performed on standard DML datasets, showing that existing deep metric learning models are vulnerable to adversarial attacks and that the proposed training protocol improves their robustness.

Positive points:
- The topic of rendering deep metric learning robust to adversarial attacks has received little attention in the past, and the results presented in the paper seem indeed novel.
- The proposed attack and robust training procedure show results consistent with the original PGD attack and the adversarial training based on it.
- The experiments show clear robustness improvements with the proposed training strategy (although, the performance of the models seems too low for practical use).
- The code for the submission is provided.

Concerns:
- The derivation of the proposed robust training framework for DML is a bit unclear. The final optimization objective seems to be stated without much justification as to why it enforces robustness.
- It would be great to be able to compare the proposed strategy with other results. Maybe R@1 could be compared to [Mao et al.] (cited by the paper)?.

Reasons for score:

Overall, I lean towards accepting the paper, as it seems to propose the first results around robust deep metric learning. The topic does seem somewhat relevant to the community. Good practices from the community around strong attacks and adversarial training seem to have been followed.

Questions / suggestions:
- How come robust performance on clean data is almost as good as baseline performance (Appendix Tab. 3)?

Minor comments:
- A few typos remain throughout the paper.
- Different values of the attack strength $\epsilon$ do not represent different threat models.
- Appendix Tab. 3: the bold value for CUB200-2011 with contrastive loss does not seem to reflect the best performance.

References:
[Mao et al.] Metric learning for adversarial robustness, 2019.

---

> ### Author Response · Authors · 2020-11-15
> **Initial response**
>
> We are thankful for your feedback and the fact that you found our method novel.
>
> Regarding the comparison to the work of Mao et al, [quoting our response to Reviewer 3]: `Mao et al. combines metric losses and adversarial robustness for a fundamentally different setting. Concretely, Mao et al. propose to "regularize the representation space under attack with metric learning to produce more robust classifiers.". Here, "representation space under attack" is internal representations of the given classifier. Comparing to their technique proves challenging as they are not performing zero-shot learning, unlike our setting which is a zero-shot learning scenario. Likewise, comparisons in the space of "traditional" robust classifiers seems non-trivial.`
>
> Could you expand upon how a suitable comparison could be established?

---

> > ### Comment · AnonReviewer4 · 2020-11-23
> > **Re: Initial response**
> >
> > Thank you for your response. As per Reviewer 3's suggestion, comparison for classification seems relevant. By evaluating the proposed method using $R@1$, a classifier is already selected implicitly in the form of the nearest neighbour. Different classifiers can be used to replace that, which in turn can be compared against other robust classifiers.

---

> > > ### Author Response · Authors · 2020-11-24
> > > **comparison with Mao et al.**
> > >
> > > We are still puzzled by this comment. In DML, first an example $x$ is embedded in a smaller dimensional space (by applying a function $f$).  Then at inference time, in the embedding space, the closest anchor is chosen among the set of anchors, achieved by comparing the distances between $f(x)$ with all anchors in the embedding space. Classification is a different task and cannot directly apply here. In Reviewer 3's suggestion, it is mentioned that perturbing the anchor is chosen in Mao et al.[2019], instead of the positive example. We chose the positive example because the attacker can only perturb the positive example at test time, and the defense should align with the attacker's capability. We also addressed this issue in our revised paper. We found that perturbing anchor points can improve robustness but its training is more unstable.
> > >
> > > If this has not addressed the comment regarding the comparison with Mao et al.[2019], could the reviewer provide a concrete example of the comparison?

---

> ### Author Response · Authors · 2020-11-24
> **Summary of revision**
>
> Throughout our revision, we have improved upon the following related to your feedback:
>
> - We have provided additional evaluations using other attack methods (C&W, FGSM) and include the performance in Appendix (see Table 3 and Table 4).
> - Updated tables, and the bold markings.
>
> *We have revised our implementation, thus experiments has been rerun and numbers are updated throughout the paper.*

---

### Official Review · AnonReviewer2 · 2020-10-28
**The idea is interesting but the paper lacks some analysis**

**Rating:** 5
**Confidence:** 3

**Review:**

edit after rebuttal:

My opinion about the paper has not changed. Although the general idea is interesting, my main concern is that the approach aims at performing defense against a specific attack. The robustness of the approach w.r.t. other attacks (such as L_2 and L_0) needs to be evaluated.

====

The paper proposes a robust deep metric learning approach to adversarial attacks. Unlike previous Deep Metric Learning (DML) approaches, the approach focuses on robust optimization-based training that uses a saddle-point formulation. The approach considers the dependence of two classic metric losses (constrastive and triplet-loss) on the samples of a mini-batch to produce adversarial attacks.
Given a pair of samples in a mini-batch i and j, a perturbed variant of i wrt j is created as formulated in the first equation of Section 3.4. Depending on whether i and j are similar or not, a perturbation delta in some epsilon-ball increases or decreases the squared Euclidean distance between the representations of i and j.

The paper then evaluates how classic deep metric learning approaches are robust to Projected Gradient Descent (PGD) attacks. The results on some standard datasets show that classic metric learning approaches are very weak to PGD attacks (see Table 1), and the proposed approach is more robust as illustrated in Table 2.

The paper is well written in general although the experimental section is sometimes hard to follow. It took me some time to understand what the difference between the tables was.
My main concern is that only one kind of adversarial attack is considered in the paper. In the second contribution, the authors state that classic DML approaches "do not have any robustness — their accuracy drops to close to zero when subjected to PGD attacks that we formulate." Classic DML approach are weak to the evaluated adversarial attack, but what about other kinds of adversarial attacks?
It is not surprising that a specific method optimized for this attack is more robust. Is the proposed approach robust to other kinds of adversarial attacks?

If I understand correctly, according to the definition of the distance in Section 3.1, the perturbation delta is performed in the input space of the neural network. How is the argmax/argmin problem solved in Section 3.4? I tried to check the code but only saw inputs perturbed by -epsilon or +epsilon and then clipped. I do not see where the argmax/argmin problem is solved, if the neural network is highly nonconvex, the problem might be hard to solve. This needs a discussion, or at least a reference.
For instance, if (x_j, x_i) are dissimilar and x_j is in the epsilon-ball centered at x_i, then rho(x_i, x_j) should be equal to x_j. How can the proposed approach be robust to such a case?

How does the proposed approach have an impact on the norm of the learned representations?
Can the authors perform an analysis on the difference of representations between classic DML and the proposed approach?
What about the robustness for a different value of epsilon or type of norm used during training?




Minor comment: The equation in Formulation 1 is confusing because i is used as index twice (once in the sum, and once in the max). Also please keep equation indices for most equations, it makes reviewing easier.

---

> ### Author Response · Authors · 2020-11-19
> **Clarification**
>
> On behalf of all the authors, we are thankful for the valuable feedback. We appreciate you find the paper well-written.
>
> *Using the $P_n$ notation for referencing paragraphs, where $n$ is the paragraph being referenced.*
>
> ($P_3$)  We choose to use PGD for its empirically-known ability to create effective adversarial perturbations, and is often seen as *the attack method to beat* [Wong & Rice et al.]. We have experimented with other (often considered weaker) attacks, such as FGSM and C&W. For FGSM we see comparable effects to robustness $\text{R@1} \leq 1.0$ and $\text{mAP@R} \leq 2.6$ for naturally-trained DML models. We have experienced problems finding suitable hyper-parameters to yield C&W effective under some reasonable running time, when compared to the runtime of PGD. However, this low efficiency is a known problem for the C&W attack, and [the author of C&W  (Carlini) even discouraged the use of the C&W attack for the $\ell_{\infty}$ norm now](https://github.com/tensorflow/cleverhans/issues/978#issuecomment-464594668) (as PGD has become available). **Is there any other attack method, that you particularly you would find relevant beyond the two we mentioned for comparison?**
>
> ($P_4$) We solve the argmin using ADAM, as suggested by Roth et al., the argmax through use of PGD [Madry et al.]. The $-\epsilon, \epsilon$  you refer to stem from the random seeding step of PGD, that randomly samples a point within the $\epsilon$-ball, and the clipping stem from "projection" (within the $\epsilon$-ball) that control the perturbed data point cannot escape the $\epsilon$-ball. We agree that solving this problem for highly non-convex neural networks, in an optimal fashion, is difficult. However, PGD have empirically been efficient at finding effective attacks for highly non-convex networks. The philosophical case of obtaining robustness for two data points ($x_i,x_j$) with intersecting $\epsilon$-balls is an interesting problem. However, this is a problem tied to limitations of $\epsilon$-ball robustness that go beyond the field of application within our work (deep metric learning). Achieving $\epsilon$-ball robustness is (generally) done under the assumption that $\epsilon$-ball of opposing classes do not intersect, as it invalidates the ability to obtain robustness (as you hinted towards). This problem is often addressed by choosing $\epsilon$ to be small (in respect to the application domain), such that this occurrence improbable.
>
>
> **References**
>
> [Wong & Rice et al.]: ["Fast is better than free: Revisiting adversarial training"](https://arxiv.org/abs/2001.03994)
>
> [Madry et al.]: ["Towards Deep Learning Models Resistant to Adversarial Attacks"](https://arxiv.org/abs/1706.06083)

---

> > ### Comment · AnonReviewer2 · 2020-11-24
> > **Thank you for the clarifications**
> >
> > P_4: Thank you for the clarification. My concern was mainly that the optima seemed difficult to find in general. Now I understand the obtained solution is approximate although correct in most cases in practice. A discussion about this should have been in the paper.
> >
> > P_3: Did you also include L_2 and L_0 attacks just like in C&W (Carlini)? I do not see them in the manuscript.

---

> > > ### Author Response · Authors · 2020-11-24
> > > **Response to: Thank you for the clarifications**
> > >
> > > ($P_4$) We are thankful for your feedback, so it allows us to improve the reading experience. This methodology is common practice within the field [Goodfellow et al., Biggio et al.]. We have now emphasized more strongly in the paper the found adversarial perturbations are approximation to the argmax.
> > >
> > > ($P_3$) We have only examined $\ell_{\infty}$, however the method is applicable to other common norms used within robust optimization ($\ell_0$, $\ell_1$, and $\ell_2$). We chose $\ell_\infty$ as it is widely used within the literature (see [robust-ml.org](https://www.robust-ml.org/defenses/)).  However, as the attack algorithm (now more explicitly covered in the Section 3.4), can be used with common attack methods within adversarial machine learning (PGD, C&W, FGSM, and others), it is not limited to the $\ell_\infty$ norm. We include performance measures  for C&W and FGSM (both across naturally-trained DML models and our robustly trained models) in Table 3, Table 4 and Table 7 under $\ell_\infty$. Besides, training a robust model resistant to multiple attacks (even only different $\ell_p$ attacks) is known hard and remains an active research area [Tramèr and Boneh]. We consider it beyond the scope of this work.
> > >
> > > **References**
> > >
> > > [Goodfellow et al.]: ["Explaining and Harnessing Adversarial Examples"](https://research.google/pubs/pub43405/)
> > >
> > > [Biggio et al.]: ["Wild Patterns: Ten Years After the Rise of Adversarial Machine Learning"](https://arxiv.org/abs/1712.03141)
> > >
> > > [Florian Tramèr, Dan Boneh]["Adversarial Training and Robustness for Multiple Perturbations"](https://arxiv.org/abs/1904.13000)

---

> ### Author Response · Authors · 2020-11-24
> **Summary of revision**
>
> Throughout our revision, we have improved upon the following related to your feedback:
>
> - Tables 1 & 2 have been updated to enhance clarity.
> - We have added performance measures for robustness for naturally-trained DML models in Appendix (see Table 7).
> - We have expanded upon our robustness evaluation, to cover also include other attack methods (C&W, FGSM). Importantly for C&W, the cost for conducting attacks forced us to use 50 iterations. (see Table 3 and Table 4 in Appendix)
> - A clarification of how the argmax is solved is added to the paper (Section 3.4, Section 3.5).
> - We have included an experiment on high-dimensional synthetic data (embedding size = 2, for visualization purposes), highlighting how the robust training objective affects the learned embedding space. See Appendix E.
> - Formulation 1 has been changed as suggested, to provide more clarity.
>
> *We have revised our implementation, thus experiments has been rerun and numbers are updated throughout the paper.*
>
> We would like to thank the reviewer for the valuable feedback that lead us to these adjustments.

---

### Official Review · AnonReviewer3 · 2020-10-29
**Learn robust representations with perturbation on reference points**

**Rating:** 4
**Confidence:** 4

**Review:**

Authors research the problem of robust metric learning in this work. They propose a min-max formulation to learn the adversarial example and robust representations, simultaneously. The empirical study confirms the effectiveness of the proposed method. My concerns are as follows.

1.	In this work, authors adopt the positive example rather than anchor in a triplet for perturbation. However, as they illustrated in Section 3.2, the attack only can be applied on anchors while the reference points are fixed. The reason for current choice is that having perturbation on positive example achieves best performance as shown in appendix. But it may be due to the problem in the algorithm or implementation since the behavior of optimizing anchors is weird as reported. Moreover, perturbing anchors work well as reported in other work [2]. The current setting for perturbation is inconsistent with the practical applications.
2.	It is not clear how $\rho(x_i, x_j)$ is obtained. Is the optimization problem solved to be optimum or just an approximated solution?
3.	There is no baseline from robust optimization for comparison. At least robust triplet loss in [2] can be included. Besides, classification is a strong baseline for metric learning [3]. Therefore, a robust classification model can be involved in the empirical study.


[1] A. Sinha, et al. Certifying Some Distributional Robustness with Principled Adversarial Training.
[2] C. Mao, et al. Metric Learning for Adversarial Robustness.
[3] A. Zhai, et al. Classification is a Strong Baseline for Deep Metric Learning.

---

> ### Author Response · Authors · 2020-11-13
> **Initial response**
>
> Thanks for the review.
>
> 1. When you state "The current setting for perturbation is inconsistent with the practical applications", it is not clear what practical applications you are referring to. Can you please clarify?
>
> 3. Despite similar naming, Mao et al. combines metric losses and adversarial robustness for a fundamentally different setting. Concretely, Mao et al. propose to "regularize the representation space under attack with metric learning to produce more robust classifiers.". Here, "representation space under attack" is internal representations of the given classifier. Comparing to their techniques proves challenging as they are not performing zero-shot learning, unlike our setting which is a zero-shot learning scenario. Likewise, comparisons in the space of "traditional" robust classifiers seems non-trivial. Could you elaborate on how you'd find this a suitable comparison?

---

> > ### Author Response · Authors · 2020-11-23
> > **Response with revision.**
> >
> > We have revised the paper and implementation to address your concerns. Consequently, results showcased in experiments have also been updated to align with the new implementation.
> >
> > 1. Mao et al.'s application of triplet loss is effectively to use it as a weighted regularization (weighted by $\lambda_1$) in combinatorial loss function (see Equation 1 in their publication) for which anchors are adversarial perturbations approximated by maximizing the cross-entropy loss for the classifier undergoing training. To emphasize the key differences to our method: (1) they seek to improve the robustness of traditional classifiers, (2) adversarial perturbations are discovered using cross-entropy loss for the respective classifier, and (3) triplet loss is one of three weighted terms in their loss function for training. We find that the conclusion of `perturbing anchors work well as reported in other work` to be an overstatement, highlighted by the key differences. After rerunning the experiments, after updates to the implementation, Appendix F includes experiments that show robustness can be obtained by perturbing anchor points. However, the experiment suggest that it is more unstable (during training) than the proposed method. We discuss this further in the revised discussion section.
> >
> > 2. Thanks for pointing this uncertainty. It is obtained using PGD (and is similarly obtainable by other common attack methods, C&W, FGSM, etc.). We have emphasized this more strongly in the revision.
> >
> > 3. We have highlighted more strongly that the typical application domain for DML, namely zero-shot learning. Thus, conducting comparisons to traditional classifiers, where the set of classes are identical at train and test time, seems infeasible as this premise does not hold for our application.

---

### Decision · Program_Chairs · 2021-01-07
**Final Decision**

**Decision:**

Reject

**Comment:**

This paper proposed a novel Adversarial Deep Metric Learning approaches. The reviews pointed out the paper proposes an interesting idea and it is among the rare works that address directly robust metric learning which an important topic for efficient metric learning.
Some concerns were raised about the analysis and the lack of comparisons notably with other types of adversarial attacks.
The authors provide a rebuttal where they addressed some concerns raised by reviewers with some precisions on the work, its positioning with respect to other related papers and additional comparisons notably with other types of attacks.
A minor remark: there is a typo in Eq(13), where the $z$ in the loss function is actually not defined and should be included in the max function.
That being said, the contribution is still limited in considering only the infinite norm, analysis and comparisons to prior work remain weak. The paper does not meet the requirements for acceptance to ICLR in its current form.
I have then to propose rejection.